# Analytical Approaches in Official Food Safety Control: An LC-Orbitrap-HRMS Screening Method for the Multiresidue Determination of Antibiotics in Cow, Sheep, and Goat Milk

**DOI:** 10.3390/molecules27196162

**Published:** 2022-09-20

**Authors:** Severyn Salis, Nicola Rubattu, Federica Rubattu, Maurizio Cossu, Andrea Sanna, Giannina Chessa

**Affiliations:** Istituto Zooprofilattico Sperimentale della Sardegna, Via Duca degli Abruzzi 8, 07100 Sassari, Italy

**Keywords:** antibiotics, milk, screening method, LC-HRMS, method validation, CIR. 2021/808

## Abstract

The presence of unauthorized substances, such as residues of veterinary medicines or chemical contaminants, in food can represent a possible health concern. For this reason, a complete legislative framework has been established in the European Union (EU), which defines the maximum limits allowed in food and carries out surveillance programs to control the presence of these substances. Official food control laboratories, in order to ensure a high level of consumer protection, must respond to the challenge of improving and harmonizing the performance of the analytical methods used for the analysis of residues of authorized, unauthorized, or prohibited pharmacologically active substances. Laboratories must also consider the state of the art of the analytical methodologies and the performance requirements of current legislation. The aim of this work was to develop a multiresidue method for the determination of antibiotics in milk, compliant with the criteria and procedures established by Commission Implementing Regulation (EU) 2021/808. The method uses an LC-Orbitrap-HRMS for the determination of 57 molecules of antibiotic and active antibacterial substances belonging to different chemical classes (beta-lactams, tetracyclines, sulfonamides, quinolones, pleuromutilins, macrolides, and lincosamides) in bovine, ovine, and goat milk samples. It provides a simple and quick sample pretreatment and a subsequent identification phase of analytes, at concentrations equal to or lower than the maximum residual limit (MRL), in compliance with Commission Regulation (EU) 2010/37. The validation parameters: selectivity, stability, applicability, and detection capability (ccβ), are in agreement with the requirements of Commission Implementing Regulation (EU) 2021/808 and demonstrated the effectiveness of the method in detecting veterinary drug residues at the target screening concentration (at the MRL level or below), with a false positive rate of less than 5%. This method represents an effective solution for detecting antibiotics in milk, which can be successfully applied in routine analyses for official food control plans.

## 1. Introduction

In EU states, the use of antimicrobial veterinary drugs is only allowed for therapeutic treatment and/or prevention of infectious diseases, while use for the auxinic purposes, as growth promoters in animal feed, has been banned since 1 January 2006 [1]. Despite the legislative regulation of these active ingredients, they often are not used appropriately, and excessive, incorrect, and sometimes fraudulent use of these drugs on farms could lead to the presence of residues in food, with potential negative impacts on human health [2,3,4,5]. The contamination of food with residues of veterinary drugs, indeed, can give rise to allergic reactions in hypersensitive subjects, alterations of the intestinal bacterial flora, and can contribute to antimicrobial resistance phenomena in some bacterial strains, which is considered an increasing threat to public health [6,7].

For these reasons, since the 1990s the European Union has issued regulatory provisions on food safety [8], asking the Member States to control antimicrobial residues thereof in various foods with animal products. Most of these compounds are regulated based upon a maximum residue limit (MRL) laid down in Commission Regulation (EU) No. 2010/37 [9]. For this purpose, the UE Commission Decision 2002/657 [10] and the more recent Commission Implementing Regulation (EU) 2021/808 [11] defined measures to monitor certain substances and established performance of analytical methods for residues of pharmacologically-active substances used in food-producing animals, and for the interpretation of results, as well as on the methods to be used in the testing of official samples.

Among animal products, milk represents an important portion of the world’s productive economy, due to the high direct consumption of this food and the variety of products obtained from its transformation. Among the most used antibiotics, mainly for mastitis treatment, are tetracyclines, sulfonamides, β-lactams (penicillins and cephalosporins), and macrolides; these drugs can become residues as pharmacologically-active substances in milk, compromising its safety and quality. The MRLs allowed for antibiotics in milk have a very wide range: 4 µg/kg for penicillins, 20 to 100 µg/kg for cephalosporins, 100 µg/kg for sulfonamides, and 40 to 200 µg/kg for macrolides. This variability, and the need to investigate extremely low concentrations, even if only in screening analyses, requires the use high-performance analytical procedures and techniques for sensitivity, specificity, and robustness, as indicated in CIR (EU) 2021/808.

Milk is a chemically complex food matrix, it contains essential nutritional components, such as significant amounts of saturated fat, protein (≈3%), and calcium, in different proportions, depending on the milk type (bovine, ovine, or goat). These components can interfere with the analytical response: proteins, for example, easily bind to some antibacterials; or divalent cations, such as calcium, can form complexes with tetracyclines, favoring their retention in different matrices [12]. For these reasons, particular attention is required in the sample preparation, extraction, and purification phases, which are often the major challenges of the method.

Different analytical strategies for milk treatment are available in the literature, for either screening or quantification of veterinary drug residues. These procedures involve protein precipitation [13], solid phase extraction (SPE) [14,15], or a quick easy cheap effective rugged safe (QuEChERS) approach [16,17]. Recently, the Waters Corporation patented a new HLB absorbent called PRiME [18,19] (process, robustness, improvements, matrix effects, ease of use), with specific adsorption characteristics for the lipid chains of dehydrated fatty acids. With HLB PRiME technology, it is possible to choose to load the sample directly into the absorbent column without prior activation, conditioning, and washing, which produces greater effectiveness for the sample-adsorbent interaction [19,20].

Liquid chromatography tandem mass spectrometry (LC-MS/MS) is the most appropriate technique available today for the analysis of a wide range of antibiotics in food, including milk [21,22,23,24,25]. Furthermore, the use of HRMS instrumentation, operating in full-scan acquisition mode to an accurate mass, offers even greater possibilities for multiresidue analysis of antibiotics, as well as for the identification of unknown compounds using non-target analysis. However, these methods were mainly developed to quantify drug residues in bovine milk [21,22,23,24,25], very few analytical procedures are available for sheep and goat milk. In some regions, sheep and goat milk have a considerable production, distribution, consumption, and use, especially for typical products of the supply chain, such as pecorino romano.

The aim of this work was to develop an LC-HRMS analytical strategy for the screening of 57 antimicrobial veterinary drug residues in the milk of bovines, ovines, and goats. A method of sample preparation was developed, in order to obtain a really simple and fast extraction step. The whole method was validated according to the requirements of CIR (EU) 2021/808. The method reported in this manuscript was applied in the laboratory for the analysis of drug residues of the Zooprophylactic Institute of Sardinia for the routine screening control of milk samples collected for the National Plan of Residues Control in animal food products.

## 2. Results

### 2.1. Method Validation

Commission Implementing Regulation (EU) 2021/808, which repeals Commission Decision 2002/657/EC, defines screening methods and those analytical techniques for which it can be demonstrated, in a documented and traceable manner, are validated and have a false compliance rate <5% (β error) at the interest level. If a non-compliant result is suspected, this result must be ascertained with a confirmatory method. Furthermore, these methods should be easily applicable to a large number of samples and designed to avoid false negative results. CIR (EU) 2021/808, as well as the previous CD 2002/657, establishes that the parameters for validating the acceptability of semi-quantitative screening methods are:Detection capability, ccβSelectivity-SpecificityStabilityRobustness

The basic principles for the evaluation/calculation of these parameters are detailed in the Guidelines for the validation of screening methods for residues of veterinary medicinal products, developed by the European Community Reference Laboratories (CRL 2010) [26]. This document is the result of the workshop organized in 2005 by the EURL, to define and use a single guide on the validation of screening methods, favoring their application for all types of substances in all matrices. Therefore, the validation procedure of this work was performed according to the CRL 2010 guidelines, combining the experiments in such a way as to allow verification of all the expected performance parameters at a statistically significant level. The above is described in the validation plan summarized in Table 1.

### 2.2. Detection Capability (ccβ)

Detection capability (ccβ) is the smallest analyte content that can be detected or quantified in a sample with an error of β. Depending on the possible regulatory limit, for each substance or class of them, the CCβ can be one of the following:(a)the lowest concentration reasonably achievable for the detection of samples containing residues, for banned or unauthorized substances(b)below the permitted limit (MRL), for authorized substances

Regarding the substances analyzed in this work, the CCβ was assessed at the screening target concentration (STC) established on the basis of the MRLs indicated in the EU Regulation 2010/37, as described in Table 2. In particular, STC corresponds to the MRL or a fraction of the MRL for permitted substances, while for those without a MRL, it was set at the lowest concentration achievable for the substance or class.

### 2.3. Specificity

The specificity of an analytical method is its power of discrimination between the analyte and any closely related substances. This parameter was assessed by statistical evaluation of independent tests, in accordance to CIR 2021/808, and carried out on 25 blank samples, representative of the types of milk (bovine, ovine, and goat) provided by the National Residue Control Plan. The absence of any signal interference, peaks, or ion traces in the retention time region of the target analyte was checked in chromatograms. Furthermore, the use of high resolution mass spectrometry (HRMS) made it possible to attribute a purely instrumental specificity to the analytical method, respecting the following parameters:–exact mass accuracy ≤5 ppm–tolerance range of retention times associated to an exact mass ≤2.5%,
which guarantees the necessary fitness for purpose in terms of selectivity.

Determination of Threshold Value (T) and Cut-Off Factor (Fm)

The analytical response of the blank samples analyzed in the specificity study, expressed in terms of concentration (µg/kg), was used to calculate the threshold value (T), corresponding to the minimum concentration of analyte above which a sample was considered positive. T was calculated according to Equation (1)
T = B + 1.64 × SB(1)
where B and SB are the mean and standard deviation of the blank sample concentrations at the retention time of each analyte, respectively.

For calculation of the cut-off factor (Fm), we used the same 25 STC fortified blank samples, calculating the mean and standard deviation of the concentrations obtained and applying the following formula:Fm = M − 1.64 × SD(2)
where M is the mean and SD is the standard deviation of the analyte concentrations in TSC-fortified samples.

Fm represents the concentration value above which must be placed 95% of the samples added to the STC. Graphically, we can describe the T and Fm parameters in the following Figure 1:

The comparison between the T and Fm values leads to the evaluation of ccβ, with two possible scenarios:Fm > T: is the optimal condition, corresponding to a percentage of false negatives less than 5%; therefore, the CCβ is less than the concentration of STC (less than or equal to MRL).Fm < T: the percentage of false negatives is greater than 5%, CCβ is greater than STC, and it is necessary to proceed with new experiments to determine the new ccβ.

In the present study, the optimal situation (Fm > T) was always verified and ccβ was defined at the target screening concentrations indicated in Table 2. The cut-off factor obtained for each analyte represents its experimental detection capacity (real ccβ), i.e., the concentration value above which a sample must be considered positive for screening and therefore must be sent for confirmation analysis. Figure 2 shows for some molecules (each representative of the chemical class to which they belong) the graphs of Fm and T obtained from the experimental data points.

The cut-off factor Fm variability was evaluated with blank milk samples fortified at the STC. The coefficient of variation (CV), under conditions of intralaboratory reproducibility (CV_R_), was compared with the CV calculated using the Horwitz equation:CV = 2 ^(1 − 0.5 log C)^(3)
where C is the mass fraction expressed as a power of 10.

As also reported in Reg. 2021/808, for mass fractions lower than 120 µg/kg, the maximum allowed coefficient of variation (%) is 25%, while for mass fractions below 10 µg/kg the allowed CV is 30%; however, it is explicitly stated that these are guide values, and at the same time it is recommended that the CV% be as low as reasonably possible.

In the present work the CV values calculated for all analytes were well below the maximum values described above, and are indicated in Table 3.

### 2.4. Stability

The standard solutions of the analytes, divided by chemical class, were stored for a period of time and under the conditions indicated by the stability studies reported by Berendsen et al. [27] and monitored with routine application of the method. Routine analysis was performed using a five-point standard solution calibration curve (0.1, 0.5, 1.0, 5.0, 10.0 µg/L) prepared in the same solvent used for the dilution of milk extracts: AcOONH4 0.2 M:MeOH 9:1 (*v*/*v*). The stability of the analytes was evaluated by verifying the linearity of the calibration line, with R^2^ ≥ 0.999, and the constancy of the intercept/slope ratio (y/x), which must fall within a tolerance range of ≤20%

### 2.5. Ruggedness

Ruggedness studies used the deliberate introduction of reasonable minor variations that may occur in a laboratory into the procedure and observation of their effects. In the present work, four potentially critical factors were identified: the temperature and speed of the centrifuge, HLB prime conditioning, amount of EDTA in the extraction phase. These four factors were tested by analyzing 10 different blank samples and with 10 different types of milk being added to the STC, and running the studies on two discrete levels of each factor (f and F), on different days with different trained operators. The application of a t-test to results of the studies demonstrated that these factors were not statistically significant, confirming the assessment of the detection capacity and specificity for all analytes.

### 2.6. Quality Control

#### 2.6.1. Internal Quality Control

Four labeled internal standards (sulfanilamide-13C_6_, cefadroxyl-d4, enrofloxacin-d5 and penicillin G-d7) were added prior to extraction to all samples (10 μg/kg), with the aim ofensuring the quality of daily results,monitoring the efficiency of the extraction procedurechecking for changes in retention times.

An additional on-going quality control was performed by inserting into each batch of analysis, together with the routine samples, both “negative control” (blank matrix) and “screen positive control” milk samples (spiked at the screening target concentration).

#### 2.6.2. External Quality Control

The developed method was tested for its screening capability by participation in proficiency tests organized by Fapas (Food and Environment Research Agency, Sand Hutton, York, UK) and Progetto Trieste (Test Veritas, Padova, Italy). Our laboratory characterized all samples correctly (compliant/suspect), without any false-negative results for all compounds included in the method. The involved classes were cephalosporins, penicillins, quinolones, and tetracyclines.

## 3. Discussion

The use of LC-MSMS techniques in the context of official food safety monitoring relates almost exclusively to the development of multiclass methods for the analysis of a wide variety of drugs in food, including antibiotics in milk [21,22,23,24,25]. The particularity in the use of these techniques, which also explains the reason for their rapid and general success, is to guarantee use of parallel innovative methodologies and highly reliable results. However, they require a substantial step in optimizing the tandem mass spectrometry (MS/MS) parameters for each compound to be analyzed (single reaction monitoring, SRM or multiple reaction monitoring MRM), and this process includes, at a minimum, the selection of precursors/product ions, as well as the optimization of the relative collision energies for the dissociation induced by the collision. This step is normally done manually, by introducing standard solutions into the mass spectrometers, which makes it time-consuming to develop an analysis method; without considering the fact that a specialized and qualified analyst is generally required to perform cross-checks between adducts, ion transitions, etc.

In addition, endogenous interference from the matrix effect of the *m/z* values can also be detected. Therefore, many factors need to be considered before reliably applying an SRM/MRM method to sample analysis.

Although, until recently, the performance of instruments operating in full MS was not comparable to that of MS/MS, in terms of dynamic range, selectivity, routine sensitivity analysis, accuracy and ease of use, today the latest generations of HR instruments, such as Orbitrap-MS, with better resolution and stability of accurate mass measurements make them extremely competitive and successfully applicable in clinical laboratories [28,29,30].

Orbitrap technology has been shown to work excellently in the quantitative analysis of a large number of pharmaceutical compounds in complex matrices, such as horse urine for doping control [31], kidney tissue and honey for toxicology and food safety [32], and rat plasma for pharmacokinetics and drug metabolism [33,34].

The full scan MS approach used in the method developed in this work with the Orbitrap mass spectrometer is based on the selectivity and sensitivity obtained from stable high resolution power (up to 70,000 at *m/z* 200) and accurate mass performance (typically <5 ppm with external calibration) on a routine basis. In addition, full scan acquisition allows virtual monitoring of all generated ion transitions, and due to the high resolution, the extracted ion chromatograms have no background when a narrow mass window is selected.

This instrumental performance, obtained in a chromatographic run of only 30 min for the screening of 57 molecules of antibiotics, highlights the practicality of the method, especially considering the fact that the sample extraction phase is very simple, fast, and effective, regardless of the origin of the milk: bovine, sheep, or goat.

Finally, as far as we know, this is the first multiresidue LC-HRMS screening method for the detection of antibiotics in milk to be validated in accordance with CIR EU 2021/808, given its very recent entry into force.

## 4. Materials and Methods

### 4.1. Chemicals and Reagents

Amoxycillin (AMOX), Penicillin V (PEN V), Penicillin G (PENG), Chloxacillin (CLOXA), Oxacillin (OXA), Dichloxacillin (DICLOXA), Ampicillin (AMP), Nafcillin (NAF), Cefazolin (CEFZ), Cephalexin (CEPL), Ceftiofur (CEFT), Cefquinome (CEFQ), Cefoperazone (CEFO), Cefapyrin (CEFA), Oxytetracycline (OTC), Doxycycline (DC), Chlortetracycline (CTC), Tetracycline (TC), 4-Epi- Oxytetracycline (4OTC), Epi –Chlortetracycline (4CTC), 4-Epi- Doxycycline (4DC), 4-Epi-Tetracycline (4TC), Valnemulin (VAL), Tiamulin (TIAM), Azithromycin (AZT), Erithromycin (ERT), Spiramycin (SPIR), Tilmicosin (TILM), Tylosin (TYL), Clindamycin (CLI), Lincomycin (LIN), Norfloxacin (NOR), Difloxacin (DIF), Sarafloxacin (SARF), Danofloxacin (DANF), Ciprofloxacin (CIPF), Marbofloxacin (MARF), Nalidisic acid (NAL), Oxolinic acid (OXO), Ofloxacin (OFLO), Flumequin (FLU), Enrofloxacin (ENF), Sulphamerazine (SLMR), Sulphaquinoxaline (SLQX), Sulphapyridine (SLPY), Sulfatiazole (SLTZ), Sulfametazine (SLMTZ), Sulfadiazin (SLDZ), Sulfamonomethoxin (SLMN), Sulfamethoxypyridazine (SLMOX) Sulfadimethoxin (SLDM), Sulfaguanidine (SLGD), Sulfaclhoropyridazine (SLCLP), Sulfanilamide (SLA), Sulfamethoxazole (SLMAZ), Sulfamethizole (SLMIZ), and Trimethoprim (TMP) standards were obtained from Sigma-Aldrich (St. Louis, MO, USA) or Dr. Ehrenstofer (LGC Group, Middlesex, UK), all with >95% certified purity. Acetonitrile (ACN) and methanol (MeOH) HPLC grade were purchased from Merck (Darmstadt, Germany) and J.T.Baker (Phillipsburg, NJ, USA), respectively. Formic acid was purchased from J.T.Baker (Phillipsburg, NJ, USA) and ammonium acetate from Sigma-Aldrich. Deionized ultra-pure water was obtained from a Milli-Q SP Reagent Water System (Millipore, Bedford, MA, US). Na_2_-EDTA was obtained from Merck (Darmstadt, Germany) and OASIS HLB Prime SPE cartridges from Waters (Milford, CT, USA). Methanol, acetonitrile, and formic acid LC-MS grade were obtained from Fisher Scientific, and Milli-Q water was produced with a Milli-Q Water Advantage System (Millipore, Billerica, MA, USA).

### 4.2. Work Solutions

Stock standard solutions of each compound were prepared at concentrations of 1000, 10 and 1 µg/mL in the suitable solvent, as specified below: methanol for TC, DC, OTC, CTC, 4CTC, 4DC, 4OTC, 4TC, SMR, SQX, SPY, SLTZ, SLMTZ, SLDZ, SLMN, SLMOX, SLDM, SLGD, SLCLP, SLA, SLMAZ, SLMIZ, TMP, VAL, AZT, ERT, TYL, TILM, CLI, LIN, NOR, DIF, SARF, DANF, MARF, OFLO, ENF, NAL, SPI, TIAM, FLU; 75:25 water: acetonitrile mixture for PENG, PENV, AMOX, AMP, CLOXA, OXA, DICLOXA, NAF, CEFZ, CEPL, CEFT, CEFQ, CEFO, CEFA; methanol with 0.2% NaOH 2M for CIPF and OXO.

These stock solutions were stored for a period of time and under the conditions indicated in the stability studies reported by Berendsen et al. [34].

The working solution at STC was prepared using the appropriate volumes of 1.0 µg/mL intermediate solutions: 40 µL for penicillins, 150 µL for penicillins 2, 250 µL for cephalosporins, 100 µL for CEFQ, 500 µL for CEFT, 500 µL for tetracyclines and epimers, 100 µL for pleuromutilins, 200 µL for macrolides, 150 µL for quinolones, 750 µL for lincosamides, and 500 µL for sulfonamides, which were diluted to a final volume of 10 mL with acetonitrile. The addition of 100 µL of this work solution to 1.0 g of milk sample was equivalent to the screening target concentration of each analyte in the sample.

### 4.3. Method Description

#### 4.3.1. Sample Preparation

The extraction procedure for milk samples consisted in adding 100 µL of EDTA 0.1 M and 4.0 mL of ACN with 2% of formic acid to 1.0 g of sample in a 25 mL polypropylene tube, then vortexing for 30 s and centrifuging at 6000 rpm at 5 °C for 10 min. Supernatant was loaded on an OASIS HLB PRiME cartridge previously conditioned with 3 mL of ACN. Finally, 100 µL of purified extract was transferred in a vial for autosampling, diluted with 900 µL of solution of ammonium acetate 0.2 M:MeOH 9:1 (*v*/*v*), and analyzed. This procedure is outlined in Figure 3 below.

#### 4.3.2. Instrumental Analysis

Liquid Chromatography

The chromatographic separation of the target compounds was carried out with a Dionex Ultimate 3000 cromatographic system, controlled by Chromeleon 7.2 Software (Thermo Fisher Scientific, Waltham, MA, USA) on a Poroshell 120 EC-C18 column (100 mm × 3.0 mm; 2.7 μm particles), preceded by a Poroshell guard column (2.1 mm × 5 mm), (Agilent Technologies, Santa Clara, CA, USA) operating at 30 °C, with a sample injection volume of 10 µL.

Mobile phase A was H_2_O ultrapure with 0.1% formic acid, and mobile phase B was MeOH. The chromatographic gradient is described in Table 4.

All antibiotics were eluted in the first 20 min of the chromatographic gradient, while the next 10 min were necessary to restore the initial conditions and return the column to stability before starting a new run.

The analysis of milk samples was performed using a 5-point calibration curve (0.1, 0.5, 1.0, 5.0, 10.0 µg/L) prepared by diluting appropriate volumes of the reference solutions of all the analytes at 10 µg/mL, until the concentrations suitable to cover the entire range of identified ccβs were obtained, as described in Table 5.

Typically, each analytical batch involved the injection, in sequence, of the standard reference solutions of the calibration curve, followed by a blank milk sample (negative control) and a fortified milk sample at screening target concentration (screen positive control), and finally the unknown milk samples.

The analysis of a blank solvent sample was inserted into the analytical sequence before and after the calibration curve, between the control samples (negative and STC) and the unknown samples, and at the end of each sequence.

High Resolution Mass Spectrometry

Mass spectrometry analysis was performed with a Q-Exactive Hybrid Quadrupole Orbitrap Mass Spectrometer (Thermo Scientific, San Jose, CA, USA) equipped with a heated electrospray ion source (HESI).

Mass spectra acquisition was carried out in full scan (FS) operating mode with a scanning mass range of 150–1000 *m/z* at the resolution power of 70,000 FWHM in positive ionization mode (ESI +). The AGC target (the number of ions in the c-Trap) was set to 3e6, with an injection time of 100 ms. FS acquisition was coupled with a targeted MS/MS experiment (dd-MS2), always in ESI+ for the same *m/z* range, with a resolving power equal to 35000 FWHM, and using the inclusion list target analytes masses and expected retention times, with a time window of 30 s. The AGC target was set to 1e6, with a maximum injection time of 100 ms. In the dd-MS2 experiment, precursor ions were filtered by the quadrupole, which operated with an isolation window of *m/z* 2.0, and fragmentation of the precursors was optimized as normalized collision energy (NCE) values (set after injection of the mix standard solution at an analyte concentration of 10 μg/L. Detection was based on the calculated exact mass of the protonated/deprotonated molecular ions, on at least two matching fragments, and on the retention times of the target compounds. The mass spectrometry device was regularly calibrated with the LTQ ESI Positive Ion Standard Solution supplied by Thermo Fisher. Table 6 summarizes the operative conditions of the Q-Exactive Orbitrap experiments.

According to CIR (EU) 2021/808, an identification point system should be used to select an appropriate acquisition method and evaluation criteria. For confirmation of the identity of substances in a matrix for which an MRL (authorized use) is established, a minimum of 4 identification points are required. Five identification points are required for unauthorized or prohibited substances.

Table 7 below indicates the identification points that can be attributed, based on the analytical technique used:

Based on these indications, all analyzed substances were identified and determined with a number of points ≥5. 

Spectrometric and chromatographic detection parameters are indicated in Table 8 for all 57 molecules, and a relative chromatogram is reported in the subsequent Figure 4.

## 5. Conclusions

A simple, sensitive, and highly specific method for screening 57 antibiotic molecules belonging to different chemical classes in milk samples was developed and validated. The method combines a fast and effective sample treatment procedure with the very high analytical sensitivity offered by the UPLC-HRMS technique. The results, in terms of the detection capability of ccβ, confirmed full compliance with the new European legislation requirements indicated by the Commission Implementing Regulation 2021/808.

The sample treatment procedure was optimized, in order to apply it indifferently on cow, goat, or sheep milk, and involved an ACN/HLB PRiME absorbent extraction (this removed any potential interference of milk lipidic component), as well as a subsequent remarkable dilution of the final extract, with the double advantage of:

avoiding the evaporation step, which is usually a critical phase regarding the potential loss of the analysis,

decreasing the matrix effect (ME), allowing the quantification of the analysis on a standard calibration curve.

This screening method is currently being applied in the routine analysis of samples of the National Residue Control Plan for monitoring veterinary drugs in food.

In addition, it could also be used for the control of non-targeted compounds, using the full spectra acquired with the HRMS full-scan mode.

## Figures and Tables

**Figure 1 molecules-27-06162-f001:**
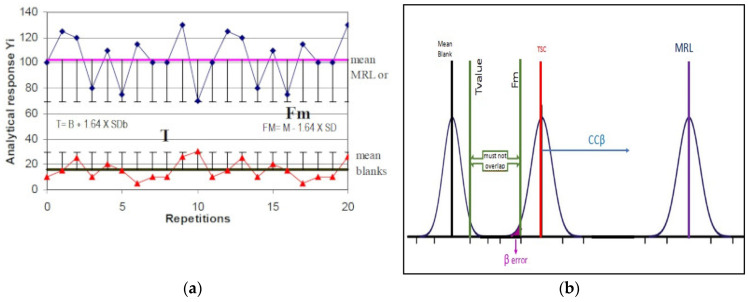
Generic graphical representation of the threshold values T and Fm, for comparison in the calculation of the ccβ: (**a**) visualization in a generic distribution of data points; (**b**) visualization in a normal distribution.

**Figure 2 molecules-27-06162-f002:**
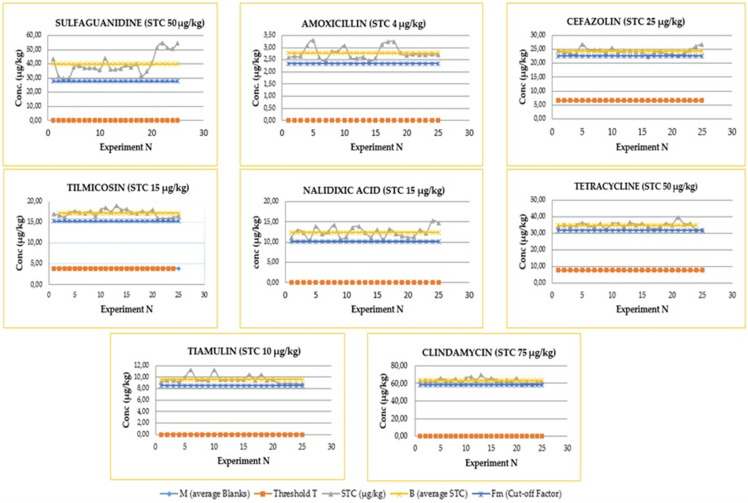
Graphical representation of the threshold and cut-off values (T and Fm) for an analyte representative of the chemical class to which it belongs.

**Figure 3 molecules-27-06162-f003:**
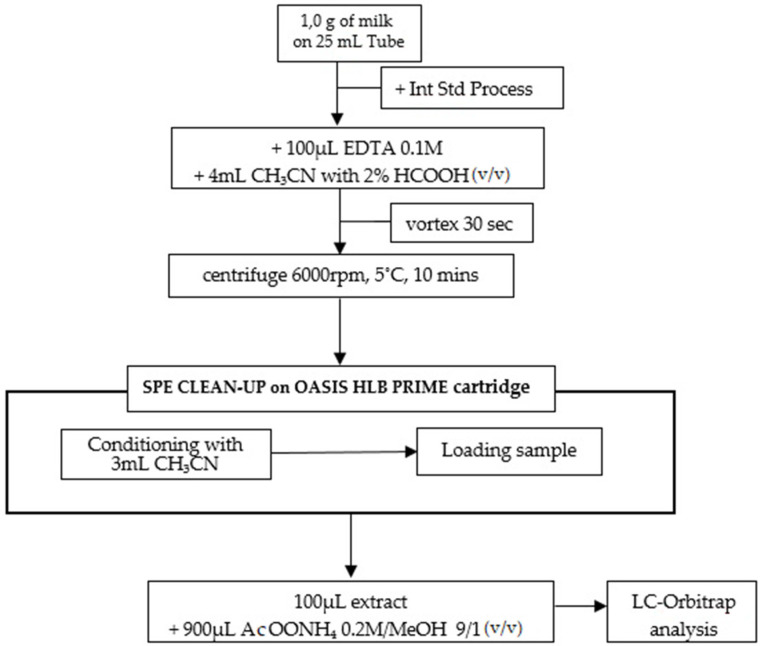
Scheme of the treatment of milk samples.

**Figure 4 molecules-27-06162-f004:**
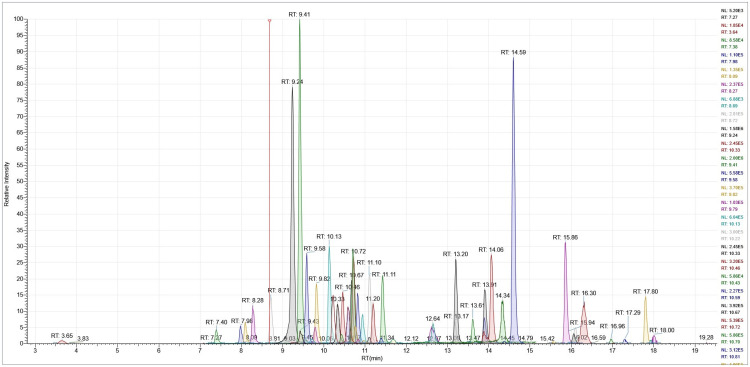
Chromatogram of a milk extract fortified at screening target concentration level.

**Table 1 molecules-27-06162-t001:** Validation plan.

Parameter	Sample Matrix	Validation Samples	Samples N	Indicator	Experiments N
**Detection Capability ccβ**	milk	10 sheep, 10 cow, 5 goat	25 spiked milk	Cutt-off factor (Fm)	25
**Specificity**	milk	10 sheep, 10 cow, 5 goat	25 blank milk	Threshold value Tv	25
**Analyte’s Stability**	Standard solution of calibration curve		5	y/x Ratio	10
**Ruggedness**	milk	Variation on 2 levels for 4 factors ^(^*^)^	10 blank + 10 STC spiked	Ccβ	20

^(^*^)^ centrifuge’s temperature and speed, conditioning HLB PRiME, amount of EDTA in extraction phase.

**Table 2 molecules-27-06162-t002:** Classes of substances, maximum residual limits (MRLs), STC for analytes (individual or by class).

Analyte	MRL (µg/kg)	STC (µg/kg)	Analyte	MRL (µg/kg)	STC (µg/kg)
Amoxicillin	4	4 ^(a)^	Lincomycin	-	75 ^(b)^
Ampicillin	4	4 ^(a)^	Oxolinic Acid	-	15 ^(b)^
Cloxacillin	30	15 ^(b)^	Ciprofloxacin	100	15 ^(b)^
Dicloxacillin	30	15 ^(b)^	Danofloxacin	30	15 ^(b)^
Nafcillin	30	15 ^(b)^	Difloxacin	-	15 ^(b)^
Oxacillin	30	15 ^(b)^	Enrofloxacin	100	15 ^(b)^
Penicillin G	4	4 ^(a)^	Flumequin	50	15 ^(b)^
Penicillin V	4	4 ^(a)^	Marbofloxacin	75	15 ^(b)^
Cefalexin	100	25 ^(b)^	Norfloxacin	-	15 ^(b)^
Cefazolin	50	25 ^(b)^	Ofloxacin	-	15 ^(b)^
Cefapirin	60	25 ^(b)^	Sarafloxacin	-	15 ^(b)^
Cefquinome	20	10 ^(b)^	Sulfaquinoxaline	100	50 ^(b)^
Cefoperazone	50	25 ^(b)^	Sulfachloropyridazine	100	50 ^(b)^
Ceftiofur	100	50 ^(b)^	Sulfadiazine	100	50 ^(b)^
Chlortetracycline	100	50 ^(b)^	Sulfadimethoxin	100	50 ^(b)^
Doxycycline	100	50 ^(b)^	Sulfaguanidine	100	50 ^(b)^
Oxytetracycline	100	50 ^(b)^	Sulfamerazine	100	50 ^(b)^
Tetracycline	100	50 ^(b)^	Sulfametazine	100	50 ^(b)^
Epi- Chlortetracycline	100	50 ^(b)^	Sulfamethizole	100	50 ^(b)^
Epi- Doxycycline	100	50 ^(b)^	Sulfamethoxazole	100	50 ^(b)^
Epi- Oxytetracycline	100	50 ^(b)^	Sulfamethoxipyridazine	100	50 ^(b)^
Epi-Tetracycline	100	50 ^(b)^	Sulfamonomethoxin	100	50 ^(b)^
Tiamulin	-	10 ^(c)^	Sulfanilamide	100	50 ^(b)^
Valnemulin	-	10 ^(c)^	Sulfapyridin	100	50 ^(b)^
Tilmicosin	-	20 ^(b)^	Sulfathiazole	100	50 ^(b)^
Tylosin	50	20 ^(b)^	Trimethoprim	50	50 ^(b)^
Azithromycin	50	20 ^(b)^			
Erythromycin	40	20 ^(b)^			
Spiramycin	200	20 ^(b)^			
Clindamycin	150	75 ^(b)^			
Nalidixic Acid	-	15 ^(b)^			

^(a)^ ccβ = MRL, ^(b)^ ccβ = fraction of MRL identified as the lowest value for substances belonging to the same class, ^(c)^ ccβ = minimum level for substances without M.

**Table 3 molecules-27-06162-t003:** Coefficient of variation under within-laboratory reproducible conditions, CV_R_ (%).

Analyte	STC (µg/kg)	CV_R_ (%)	Analyte	STC (µg/kg)	CV_R_ (%)	Analyte	STC (µg/kg)	CV_R_ (%)
Amoxicillin	4	9.7	Epi- Oxytetracycline	50	5.6	Ofloxacin	15	5.9
Ampicillin	4	7.1	Epi-Tetracycline	50	3.7	Sarafloxacin	15	3.4
Cloxacillin	15	3.6	Tiamulin	10	6.7	Sulfaquinoxaline	50	6.2
Dicloxacillin	15	6.6	Valnemulin	10	11.5	Sulfachloropyridazine	50	4.9
Nafcillin	15	7.0	Tilmicosin	20	7.0	Sulfadiazine	50	6.4
Oxacillin	15	4.9	Tylosin	20	8.6	Sulfadimethoxin	50	7.7
Penicillin G	4	14.2	Azithromycin	20	2.9	Sulfaguanidine	50	18.5
Penicillin V	4	16.2	Erythromycin	20	4.4	Sulfamerazine	50	6.4
Cefalexin	25	7.1	Spiramycin	20	4.3	Sulfametazine	50	6.0
Cefazolin	25	4.5	Clindamycin	75	4.8	Sulfamethizole	50	5.7
Cefapirin	25	6.3	Lincomycin	75	4.2	Sulfamethoxazole	50	5.0
Cefquinome	10	11.2	Nalidixic Acid	15	10.9	Sulfamethoxipyridazine	50	3.2
Cefoperazone	25	4.3	Oxolinic Acid	15	5.1	Sulfamonomethoxin	50	3.5
Ceftiofur	50	5.1	Ciprofloxacin	15	4.7	Sulfanilamide	50	6.2
Chlortetracycline	50	4.9	Danofloxacin	15	4.1	Sulfapyridin	50	3.5
Doxycycline	50	8.3	Difloxacin	15	6.0	Sulfathiazole	50	3.9
Oxytetracycline	50	7.8	Enrofloxacin	15	4.5	Trimethoprim	50	7.1
Tetracycline	50	5.3	Flumequin	15	6.4			
Epi- Chlortetracycline	50	3.3	Marbofloxacin	15	5.1			
Epi- Doxycycline	50	2.7	Norfloxacin	15	6.5			

**Table 4 molecules-27-06162-t004:** Chromatographic gradient.

Time (min)	A% Formic Acid 0.1%	B% Methanol	Flow mL/min
0	95	5	0.250
1.00	95	5	0.250
20.00	5	95	0.250
25.00	5	95	0.250
26.00	95	5	0.250
30.00	95	5	0.250

**Table 5 molecules-27-06162-t005:** Preparation of the calibration curve.

Calibration Point	Concentration Level (µg/L)	Volume of Stock Standard Solution at 10 µg/mL	Final Volume AcOONH_4_ 0.2 M:MeOH 9:1
1	0.1	10 µL	10 mL
2	0.5	50 µL	
3	1.0	100 µL	
4	5.0	500 µL	
5	10.0	1000 µL	

**Table 6 molecules-27-06162-t006:** Operative conditions of Q-Exactive Orbitrap experiments.

Full MS		dd-MS2
Resolution: 70,000	Scan Range: 150–1000 *m/z*	Resolution: 35,000
AGC Target: 3e6	Auxiliary Gas: 15	AGC target: 1e6
Maximum IT: 100 ms	Polarity: ES+	Maximum IT: 100 ms
Capillary Temperature: 300 °C	Capillary (kV): 3.0	Source temperature (°C): 320
	Sheath Gas: 35	

**Table 7 molecules-27-06162-t007:** Identification point per technique.

Technique	Identification Points
**Separation (UPLC)**	1.0
**HR-MS Precursor Ion**	1.5
**Ion Product (HR-MS^n^)**	2.5

**Table 8 molecules-27-06162-t008:** Retention times, precursor exact masses, adducts, and fragmentation products of the analytes.

Chemical Class	Analyte	Formula	Specie	RT (min)	Precursor (*m/z*)	Fragment1	Fragment2	N(CE)
Betalattamics Penicillins (8)	Amoxicillin	C_16_H_19_N_3_O_5_S	[M + H]+	7.22	366.1118	208.0	349.1	10
	Ampicillin	C_16_H_19_N_3_O_4_S	[M + H]+	11.02	350.1169	106.1	192.0	20
	Cloxacillin	C_19_H_18_ClN_3_O_5_S	[M + H]+	17.24	436.0728	277.0	160.0	10
	Dicloxacillin	C_19_H_17_Cl_2_N_3_O_5_S	[M + H]+	17.89	470.0339	160.0	311.0	15
	Nafcillin	C_21_H_22_N_2_O_5_S	[M + H]+	17.95	415.1322	199.1	256.1	20
	Oxacillin	C_19_H_19_N_3_O_5_S	[M + H]+	16.91	402.1118	160.0	243.1	15
	Penicillin G	C_16_H_18_N_2_O_4_S	[M + Na]+	15.90	357.0882	160.0	176.1	10
	Penicillin V	C_16_H_18_N_2_O_5_S	[M + Na]+	16.93	373.0829	160.0	192.1	15
Betalattamics Cephalosporins (6)	Cefalexin	C_16_H_17_N_3_O_4_S	[M + H]+	10.37	348.1013	158.0	174.1	40
	Cefazolin	C_14_H_14_N_8_O_4_S_3_	[M + H]+	10.80	455.0373	156.0	153.0	15
	Cefapirin	C_17_H_17_N_3_O_6_S_2_	[M + H]+	8.08	424.0632	152.0	292.1	25
	Cefquinome	C_23_H_24_N_6_O_5_S_2_	[M + 2H]+	8.66	265.0695	134.1	324.1	16
	Cefoperazone	C_25_H_27_N_9_O_8_S_2_	[M + H]+	11.37	646.1497	143.1	290.1	16
	Ceftiofur	C_19_H_17_N_5_O_7_S_3_	[M + H]+	13.84	524.0363	241.0	210.0	25
Tetracyclines Epi-tetrcyclines (4+4)	Chlortetracycline	C_22_H_23_ClN_2_O_8_	[M + H]+	12.55	479.1216	444.1	154.0	26
	Doxycycline	C_22_H_24_N_2_O_8_	[M + H]+	14.21	445.1621	428.1	410.1	30
	Oxytetracycline	C_22_H_24_N_2_O_9_	[M + H]+	10.53	461.1555	426.1	337.1	30
	Tetracycline	C_22_H_24_N_2_O_8_	[M + H]+	10.27	445.1605	154.0	410.1	30
	Epi- Chlortetracycline	C_22_H_23_ClN_2_O_8_	[M + H]+	11.58	479.1216	444.1	154.0	26
	Epi- Doxycycline	C_22_H_24_N_2_O_8_	[M + H]+	13.57	445.1605	428.1	410.1	30
	Epi- Oxytetracycline	C_22_H_24_N_2_O_9_	[M + H]+	10.07	461.1555	426.1	201.1	30
	Epi-Tetracycline	C_22_H_24_N_2_O_8_	[M + H]+	9.38	445.1605	410.1	392.1	30
Pleuromutilins (2)	Tiamulin	C_28_H_47_NO_4_S	[M + H]+	15.70	494.3299	192.1	119.0	30
	Valnemulin	C_31_H_52_N_2_O_5_S	[M + H]+	17.62	565.3670	263.1	164.1	30
Macrolides (5)	Tilmicosin	C_46_H_80_N_2_O_13_	[M + 2H]+	13.94	435.2903	174.1	696.5	32
	Tylosin	C_46_H_77_NO_17_	[M + H]+	15.93	916.5264	174.1	101.1	25
	Azithromycin	C_38_H_72_N_2_O_12_	[M + H]+	13.05	749.5171	158.1	83.0	28
	Erythromycin	C_37_H_67_NO_13_	[M + H]+	16.14	734.4685	158.1	83.0	20
	Spiramycin	C_43_H_74_N_2_O_14_	[M + 2H]+	12.48	422.2643	540.3	699.4	30
Lincosamides (2)	Clindamycin	C_18_H_33_ClN_2_O_5_S	[M + H]+	14.46	425.1872	126.1	377.2	30
	Lincomycin	C_18_H_34_N_2_O_6_S	[M + H]+	9.17	407.2210	126.1	359.2	30
Quinolones (11)	nalidixic Acid	C_12_H_12_N_2_O_3_	[M + H]+	15.51	233.0921	205.1	159.1	70
	oxolinic Acid	C_13_H_11_NO_5_	[M + H]+	13.86	262.0710	160.0	234.0	80
	Ciprofloxacin	C_17_H_18_FN_3_O_3_	[M + H]+	10.60	332.1405	231.1	203.1	65
	Danofloxacin	C_19_H_20_FN_3_O_3_	[M + H]+	10.75	358.1562	82.1	255.1	70
	Difloxacin	C_21_H_19_F_2_N_3_O_3_	[M + H]+	11.03	400.1467	299.1	58.1	65
	Enrofloxacin	C_19_H_22_FN_3_O_3_	[M + H]+	10.66	360.1718	203.1	245.1	60
	Flumequin	C_14_H_12_FNO_3_	[M + H]+	16.01	262.0874	238.1	220.0	80
	Marbofloxacin	C_17_H_19_FN_4_O_4_	[M + H]+	9.54	363.1463	72.1	320.1	25
	Norfloxacin	C_16_H_18_FN_3_O_3_	[M + H]+	10.40	320.1405	231.1	203.1	80
	Ofloxacin	C_18_H_20_FN_3_O_4_	[M + H]+	10.07	362.1511	261.1	221.1	50
	Sarafloxacin	C_20_H_17_F_2_N_3_O_3_	[M + H]+	11.34	386.1311	299.1	338.1	60
Sulfonamides (15)	Sulfaquinoxaline	C_14_H_12_N_4_O_2_S	[M + H]+	13.56	301.0754	156.0	108.0	38
	Sulfachloropyridazine	C_10_H_9_ClN_4_O_2_S	[M + H]+	10.75	285.0208	156.0	108.0	35
	Sulfadiazine	C_10_H_10_N_4_O_2_S	[M + H]+	7.39	251.0597	156.0	108.0	35
	Sulfadimethoxin	C_12_H_14_N_4_O_4_S	[M + H]+	13.16	311.0809	156.1	108.0	42
	Sulfaguanidine	C_7_H_10_N_4_O_2_S	[M + H]+	3.21	215.0597	156.0	108.0	40
	Sulfamerazine	C_11_H_12_N_4_O_2_S	[M + Na]+	8.69	287.0573	156.0	190.0	42
	Sulfametazine	C_12_H_14_N_4_O_2_S	[M + H]+	9.80	279.0910	124.1	156.0	42
	Sulfamethizole	C_9_H_10_N_4_O_2_S_2_	[M + H]+	9.78	271.0318	156.0	108.0	40
	Sulfamethoxazole	C_10_H_11_N_3_O_3_S	[M + H]+	10.91	254.0594	156.0	108.0	40
	S.methoxipyridazine	C_11_H_12_N_4_O_3_S	[M + H]+	10.20	281.0703	126.1	108.0	50
	Sulfamonomethoxin	C_11_H_12_N_4_O_3_S	[M + H]+	11.17	281.0710	156.0	108.0	41
	Sulfanilamide	C_6_H_8_N_2_O_2_S	[M + H][NH_3_]+	3.65	156.0114	65.0	92.0	70
	Sulfapyridin	C_11_H_11_N_3_O_2_S	[M + H]+	8.27	250.0645	156.0	184.1	43
	Sulfathiazole	C_9_H_9_N_3_O_2_S_2_	[M + H]+	7.97	256.0209	156.0	108.0	38
	Trimethoprim	C_14_H_18_N_4_O_3_	[M + H]+	9.37	291.1452	123.1	261.1	60

## Data Availability

The data used to support the findings of this study are available from the corresponding author upon request.

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
