# Peer review of "Analytical Approaches in Official Food Safety Control: An LC-Orbitrap-HRMS Screening Method for the Multiresidue Determination of Antibiotics in Cow, Sheep, and Goat Milk"

_molecules, 2022, doi:10.3390/molecules27196162_

Round 1

Reviewer 1 Report

It is opinion of the reviewer that this paper before acceptance by Molecules needs several corrections/modifications. My individual comments are listed below.

L. 40 – It should be “[2-5]”.

L. 55/56 – The antibiotics names should be written with lower case letters.

L. 70 – It should be “… strategies of milk treatment”.

L. 72 – It should be “solid phase extraction (SPE)”.

L. 81 – A references should be cited.

L. 47-95 – This part of Introduction should be divided into 3-4 paragraphs.

2.1; 2.1.1; 2.1.2 – must be shifted to “4. Material and methods”.

Discussion – The LC-Orbitrap-HRMS should be compared/discussed  with other methods used for determination of antibiotics in milk.

L. 482/483 – Journal title abbreviations is needed.

L. 500 – It should be “TrAC Trends Anal. Chem.”.

Reviewer 2 Report

The manuscript title “Analytical Approaches in Official Food Safety Control: an LC- 2 Orbitrap-HRMS Screening Method for the Multiresidue Deter- 3 mination of Antibiotics in Cow, Sheep and Goat Milk” have scientific value but methods are not explained well and need significant improvements. The author didn’t provided the analytical conditions, column configuration, systematic HPLC operating system details such as HPLC sample running protocol etc. of the HPLC. Only reagents, chemical, and working solution details are provided. My decision is to thoroughly revise the manuscript before acceptance. My other comments for authors are as follows: 

Reviewer Comments:

1-      In abstract, the background of this research is missing. The author should highlight the background, problem, and why this study needs to be done, what was the scientific questions etc. Add just 2-3 lines in the beginning of the abstract, than aim of this research, than results, and finally conclusion.

2-      In Line 167 the author mentioned figure 2 and in line 282 author again said figure 2; actually it will be figure 3. Please revise all the figure numbers in the MS.

3-       Please provide the detail running operating system of HPLC. Which is missing in the methods section: “. The author didn’t provided the analytical conditions, column configuration, systematic HPLC operating system details such as HPLC sample running protocol etc. of the HPLC. Only reagents, chemical, and working solution details are provided.”

Round 2

Reviewer 1 Report

The authors corrected this paper properly taken under considerations all my comments. Therefore, I can accept it now.

Reviewer 2 Report

the authors did sufficient revision.